# A unique cell division protein critical for the assembly of the bacterial divisome

Xiao Chu[1]*, Lidong Wang[1], Yiheng Zhu[1], Zhengshan Feng[1], Qingtian Guan[2], Lei Song[1]*, Zhaoqing Luo[1]*

[1]Department of Respiratory Medicine, Infectious Diseases and Pathogen Biology Center State Key Laboratory for Diagnosis and Treatment of Severe Zoonotic Infectious Diseases, Key Laboratory for Zoonosis Research of the Ministry of Education, The First Hospital of Jilin University, Changchun, China; [2]Bioinformatics Laboratory, The First Hospital of Jilin University, Changchun, China

**\*For correspondence:**
chuxiao@jlu.edu.cn (XC);
lsong@jlu.edu.cn (LS);
luoz@jlu.edu.cn (ZL)

**Competing interest:** The authors declare that no competing interests exist.

**Abstract** Identification of unique essential bacterial genes is important for not only the understanding of their cell biology but also the development of new antimicrobials. Here, we report a previously unrecognized core component of the *Acinetobacter baumannii* divisome. Our results reveal that the protein, termed Aeg1 interacts with multiple cell division proteins, including FtsN, which is required for components of the divisome to localize to the midcell. We demonstrate that the FtsA$_{E202K}$ and FtsB$_{E65A}$ mutants effectively bypassed the need of Aeg1 by *A. baumannii*, as did the activation variants FtsW$_{M254I}$ and FtsW$_{S274G}$. Our results suggest that Aeg1 is a cell division protein that arrives at the division site to initiate cell division by recruiting FtsN, which activates FtsQLB and FtsA to induce the septal peptidoglycan synthase FtsWI. The discovery of the new essential cell division protein has provided a new target for the development of antibacterial agents.

## eLife assessment

This **useful** study shows that the essential Acinetobacter baumannii gene Aeg1 likely plays an key role in cell division. The strength of the work is the discovery that the depletion of Aeg1 leads to cell filamentation and that gain-of-function mutations in cell division genes FtsB and FtsL rescue the lethality of Aeg1 depletion. However, Aeg1's localization pattern and its requirement for other division proteins' localizations require further characterization of the functionality of fluorescent fusion proteins, fluorescence images of higher quality, and improvements in statistic qualifications, leaving the study' evidence for Aeg1's exact role in cell division **incomplete** at this time. In conclusion, the critical role of Aeg1 in the assembly of the A. baumannii divisome has yet to be established unambiguously.

## Introduction

*Acinetobacter baumannii* is a Gram-negative bacterial pathogen that causes a wide variety of nosocomial infections (*Wong et al., 2017*; *Piperaki et al., 2019*), including urinary tract infections, gastrointestinal and skin/wound infections, and secondary meningitis (*Peleg et al., 2008*; *Nasr, 2020*; *Doi et al., 2015*). The challenge posed by this pathogen is compounded by the emergence of isolates that are resistant to almost all of the available antibiotics, including some of the last resort for treatment of infections caused by bacterial pathogens, including colistin, tigecycline, and carbapenems (*Piperaki et al., 2019*; *Karakonstantis, 2021*). As a result, the World Health Organization (WHO) has classified *A. baumannii* as a pathogen for which new antibiotics are urgently needed (*Whiteway et al., 2022*).

Meeting this challenge requires the discovery of antimicrobials of new mechanisms with action or other mitigation strategies that enable reduction of disease burden caused by this pathogen.

Historically, the discovery of antibiotics was largely based on the search for secondary metabolites produced by soil-dwelling microbes (*Clardy et al., 2006*; *Wright, 2017*). The majority of clinically used antibiotic classes originate from soil bacteria, including lactams, aminoglycosides, polymyxins, and glycopeptides (*Hutchings et al., 2019*). Unfortunately, natural products discovery is now plagued by problems of so-called 'dereplication problem', wherein rediscovery of molecules or their close relatives (*Cox et al., 2017*). As a result, there has not been a novel antibiotic discovered in over 30 years using traditional methods of screening environmental samples (*Lewis, 2020*). The overuse of these merely variants of existing compounds has further accelerated the problem associated with antibiotic resistance. Thus, identification of novel therapeutic targets in the pathogen may serve as an effective alternative to the problem.

A better understanding of gene function associated with *A. baumannii* pathogenesis and biology will provide us strategies to develop antibiotics with potentially novel mechanisms of action. One extensively explored avenue in this regard is the identification of previously unrecognized essential genes by methods enabled by new DNA sequencing technologies, including insertion sequencing (INSeq) (*Goodman et al., 2009*), transposon sequencing (Tn-seq) (*van Opijnen et al., 2009*), transposon-directed insertion site sequencing (TraDIS) (*Langridge et al., 2009*), and high-throughput insertion tracking sequencing (HITS) (*Gawronski et al., 2009*). Application of these methods has identified genes believed to be required for bacterial viability under various conditions in such important pathogens as *Streptococcus pneumoniae* (*van Opijnen et al., 2009*), *Pseudomonas aeruginosa* (*Gallagher et al., 2011*), *Vibrio cholerae* (*Dong et al., 2013*), and *A. baumannii* (*Bai et al., 2021*; *Gallagher et al., 2015*; *Wang et al., 2014*).

Similar to candidate genes identified by other methods, putative essential genes obtained by In-seq were deduced from sequencing results of surviving mutants, some of which can be false positives or negatives due to unsaturated mutant libraries or polar effects on downstream genes of the insertion sites. Using In-seq Wang et al. identified 453 candidate essential genes for the growth of *A. baumannii* strain ATCC 17978 on rich medium (*Wang et al., 2014*). Among these, genes annotated as hypotheticals with orthologs in other taxonomically distant microorganisms are of particular interest because their potential role in cellular processes that can be explored as targets for the development of novel antimicrobials. We thus set out to test a subgroup of the predicted essential genes by a conditional gene deletion strategy based on the arabinose-inducible promoter (*Guzman et al., 1995*). From 10 genes examined, three were found to be essential for *A. baumannii* viability on rich medium. Further study revealed that one of these genes codes for a novel bacterial cell division protein.

## Results

### Identification of genes required for *A. baumannii* viability

Analysis of strains of *A. baumannii*, including the model strain 17,978 by In-seq technologies have led to the identification of at least 453 genes potentially required for bacterial growth on rich medium (*Gallagher et al., 2015*; *Wang et al., 2014*). Whereas many of these genes are involved in known cellular processes fundamental for cell viability such as DNA replication, transcription, and translation, some code for proteins of unknown activity, are thus of great interest as potentially new antimicrobial targets (*Gallagher et al., 2015*). To determine the function of the candidate essential genes annotated as hypotheticals, we first confirmed their essentiality by a conditional gene deletion method. To this end, we employed a set of plasmids that allow inducible gene expression in *A. baumannii* (*Jie et al., 2021*). Each gene of interest was expressed from the arabinose-inducible promoter (*Guzman et al., 1995*), and the corresponding gene was deleted from the chromosome by the standard allele exchange method (*Merriam et al., 1997*; *Figure 1A*). Chromosomal deletion mutants harboring the arabinose-inducible gene were tested for essentiality by spotting serially diluted cells onto solid medium with or without arabinose (ara) (*Figure 1A*). If the gene of interest is essential, the bacterium should fail to grow on plates without ara.

Our initial analysis of 10 genes (*Supplementary file 1*) identified *A1S_0137*, *A1S_2249*, and *A1S_3387* (*Smith et al., 2007*) that met the criteria of being required for growth on Luria Bertani (LB) agar (*Figure 1B*). Importantly, gene *A1S_3387* is also essential for *A. baumannii* growth in the

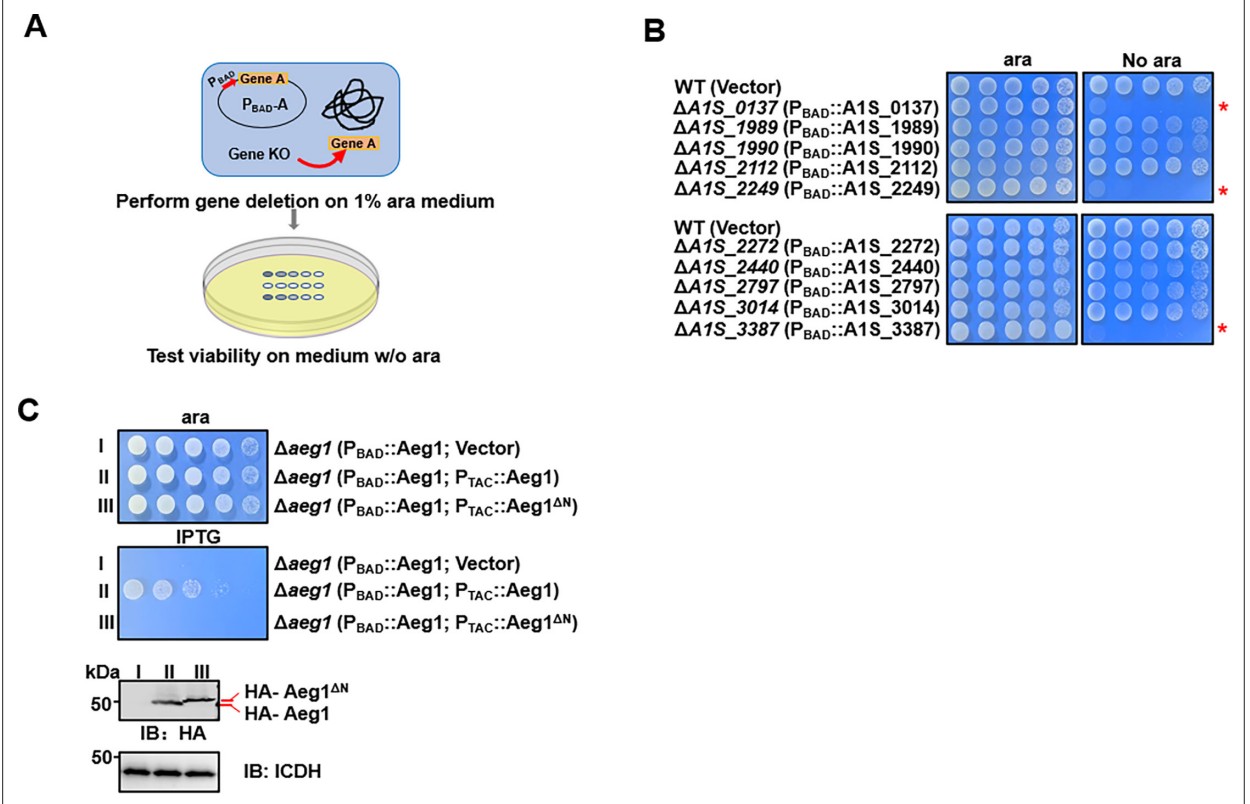

**Figure 1.** Identification of *aeg1* as an essential gene for *A. baumannii* viability on rich medium. (**A**) A diagram depicting the method for conditional deletion of potential essential genes. The gene to be examined was expressed from the arabinose-inducible promoter on a plasmid and the gene was deleted from the chromosome using the standard allele exchange method (upper panel) and strains lacking the chromosomal gene were tested for essentiality by spotting diluted cells on media with and without arabinose, respectively (lower panel). (**B**) Identification of *A1S_0137*, *A1S_2249*, and *A1S_3387* as *A. baumannii* essential genes. Each of the 10 genes predicted to be essential were deleted by the method described in A and the resulting bacterial strains were tested for growth by spotting serially diluted cells on medium with or without 1% arabinose. Images were acquired after incubation at 37°C for 18 hr. Similar results were obtained in at least three independent experiments. (**C**) Cells harboring differentially regulated HA-Aeg1 and HA-Aeg1$^{\Delta N}$ spotted onto Luria Bertani (LB) agar supplemented with 0.5 mM isopropyl-β-D-thiogalactopyranoside (IPTG) or 1% ara, images were acquired after 18 hr incubation at 37°C (C, upper two panels). The expression of the Aeg1 and Aeg1$^{\Delta N}$ were examined by immunoblotting with the HA-specific antibody. The metabolic enzyme isocitrate dehydrogenase (ICDH) was probed as a loading control (C, lower two panels). Similar results were obtained in three independent experiments. The red asterisk denotes that the deletion of this gene results in pronounced growth defects in the cells.

The online version of this article includes the following source data and figure supplement(s) for figure 1:

**Source data 1.** PDF file containing original western blots for *Figure 1C*, indicating the relevant bands and treatments.

**Source data 2.** Original files for western blot analysis displayed in *Figure 1C*.

**Figure supplement 1.** *aeg1* is essential for *A. baumannii* growth in the VBS medium.

**Figure supplement 2.** Alignment of Aeg1 homologs from a few Gram-negative bacteria.

Vogel–Bonner minimal medium with succinate (VBS) as the sole carbon source (***Vogel and Bonner, 1956***; *Figure 1—figure supplement 1*). These genes were thus designated as *A. baumannii* <u>e</u>ssential <u>g</u>enes (Aeg) and Aeg1 (*A1S_3387*) was chosen for detailed study.

Sequence analysis revealed that Aeg1 has orthologs in a number of Gram-negative bacteria, including *Klebsiella pneumoniae*, *Salmonella enterica* members of the *Moraxellaceae* family and some strains of *Escherichia coli*, with sequence similarity ranging from 80% to identical (***Figure 1—figure supplement 2***). Aeg1 is predicted to harbor a single transmembrane domain, however, the precise location of this transmembrane segment has been predicted differently across various analyses. SMART Web site (***Käll et al., 2004***) predicted the transmembrane region to be located at the N-terminus of Aeg1 (7–29 aa). In contrast, Phobius, based on HMM(HMMTOP (enzim.hu)) suggested the transmembrane segment is situated more centrally within the Aeg1 protein (134–151 aa), and further proposed that the N-terminus may correspond to a signal peptide (***Käll et al., 2007***; ***Geisinger et al.,***

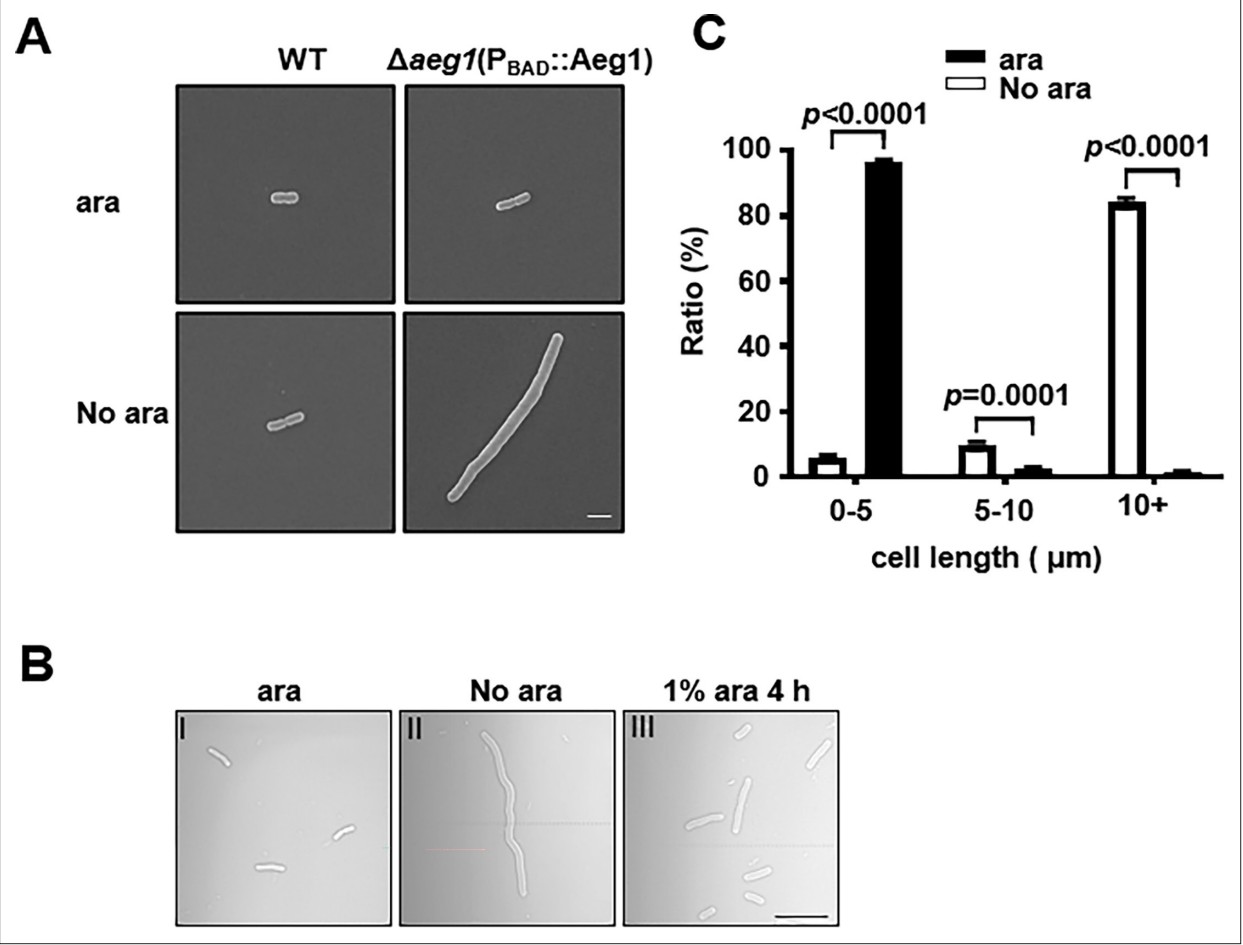

**Figure 2.** Aeg1 deficiency causes cell elongation in *A. baumannii*. (**A**) Cell morphology of the wild-type and strain Δ*aeg1*(P_BAD::Aeg1) grown in medium with or without ara. The Δ*aeg1*(P_BAD::Aeg1) and the wild-type strains were grown in medium with or without ara for 18 hr, samples were processed for scanning electron microscope imaging. Images shown are representative of three parallel cultures. Bar, 2 μm. (**B, C**) Expression of Aeg1 reversed the cell elongation phenotype caused by its depletion. Saturated bacterial cultures of Δ*aeg1*(P_BAD::Aeg1) diluted in fresh Luria Bertani (LB) broth with (I) or without (II) ara were incubated for 18 hr. The culture from the uninduced sample was split into two subcultures, ara was added to one of the subcultures and the morphology of the cells was determined at 4 hr (III) post induction. Images are representative of three parallel cultures. Bar, 10 μm (**B**). Quantitation of the cell length in samples differently expressing Aeg1. Cell length was categorized into groups of 0–5 μm, 5–10 μm, and more than 10 μm. At least 300 cells were counted for each sample (**C**). Results shown were from the average of three independent experiments. Statistical analysis in each panel was performed by Student's *t*-test.

*2020*). Regardless of the specific transmembrane topology prediction, the N-terminal region of Aeg1 appears to be a critical domain. Therefore, to further elucidate the structural and functional importance of the Aeg1 close to its N-terminus end, we employed a dual inducible system with two compatible plasmids that use promoters that respond to ara and isopropyl-β-D-thiogalactopyranoside (IPTG), respectively. Strain Δ*aeg1*(P_BAD::Aeg1, P_TAC::Aeg1) was able to grow on medium containing 1% ara or 0.5 mM IPTG (*Figure 1C*) indicating that the chromosomal deletion can be complemented by either of the constructs when expression of Aeg1 was induced. Importantly, expression of the Aeg1 mutant lacking N-terminal region was unable to restore the growth of the strain (*Figure 1C*), suggesting that N-terminal domain of Aeg1 is essential for its activity.

## Depletion of Aeg1 caused cell filamentation in *A. baumannii*

We next examined the morphology of *A. baumannii* lacking Aeg1 by optical and scanning electron microscopy. Cultures of strain Δ*aeg1*(P_BAD::Aeg1) were induced by ara for 14 hr, cells diluted into fresh LB with or without ara were allowed to grow for 18 hr. We found that cells of cultures grown in medium containing 1% ara exhibited a short rod cell shape similar to that of the wild-type strain (*Figure 2A*).

In medium without ara most cells of strain Δ*aeg1*(P$_{BAD}$::Aeg1) displayed an elongated morphology which is in sharp contrast to those of wild-type cells grew under identical conditions (*Figure 2A*). Quantitative analysis indicated that in medium supplemented with ara, more than 95% of the cells of strain Δ*aeg1*(P$_{BAD}$::Aeg1) were shorter than 5 µm. Incubation in medium without ara for 18 hr caused more than 96% of the cells to become elongated with cell lengths ranging from 5 µm to over 10 µm (*Figure 2B*). The phenotype can be partially reversed after growing the elongated cells in medium containing 1% ara for 4 hr (*Figure 2C*). These results indicate that depletion of Aeg1 causes cell elongation, a phenotype that had been observed in recent studies using transposon insertion sequencing (*Bai et al., 2021*; *Pichoff et al., 2012*).

## Mutations in the cell division gene *ftsA* bypasses the requirement of Aeg1

To explore the cellular process participated by Aeg1, we sought to isolate suppressor mutants that can survive in standard medium under conditions the gene is not expressed. To this end, cells of strain Δ*aeg1*(P$_{BAD}$::Aeg1) grown in medium supplemented with ara were exposed to the mutagen ethyl methanesulfonate (EMS), and the treated cells were plated on LB agar without ara. A few colonies were obtained from approximately $6 \times 10^8$ cells and two independent mutants M1 and M2 were chosen for further analysis. In spotting assays using diluted cells, each of these two mutants derived from strain Δ*aeg1*(P$_{BAD}$::Aeg1) was able to grow on medium without ara at rates comparable to those of wild-type bacteria (*Figure 3A*). We then performed whole-genome sequencing analysis of these two mutants, which revealed that M1 and M2 had 320 and 328 substitution mutations, respectively. In addition, 659 insertion–deletion (InDel) mutations were found in M1 and 665 such mutations were associated with M2 (*Supplementary file 2*).

Because depletion of Aeg1 caused cell elongation, we first examined mutated genes whose functions are known to be involved in cell division. An E202K mutation caused by a G to A substitution in the first position of the codon was found in the cell division protein FtsA in both M1 and M2 (*Supplementary file 2*). We thus verified the suppression phenotype by introducing a plasmid expressing FtsA or FtsA$_{E202K}$ into strain Δ*aeg1*(P$_{BAD}$::Aeg1) and examined the growth of the resulting bacterial strains under different induction conditions. Expression of FtsA$_{E202K}$ allowed strain Δ*aeg1*(P$_{BAD}$::Aeg1) to gain the ability to grow in the absence of ara. In contrast, although the protein was expressed at a comparable level, FtsA cannot suppress the growth defect caused by Aeg1 depletion (*Figure 3B*). Consistent with the growth phenotypes, expression of FtsA$_{E202K}$ in strain Δ*aeg1*(P$_{BAD}$::Aeg1) under conditions in which the complementation gene was not expressed displayed a morphology similar to that of wild-type bacteria (*Figure 3C*).

Previous studies have found that several mutations in FtsA bypassed the requirement of a number of essential cell division proteins in *E. coli*, including ZipA (*Berezuk et al., 2020*), FtsK (*Bernard et al., 2007*), and FtsN (*Geissler et al., 2003*). Among these, FtsA$_{R286W}$ (FtsA*) was the first gain-of-function mutation found in *E. coli*, which allowed the survival of mutants lacking ZipA (Δ*zipA*) or FtsK (Δ*ftsK*) (*Berezuk et al., 2020*; *Bernard et al., 2007*; *Liu et al., 2015*). Additionally, FtsA$_{I143L}$(*Park et al., 2021*) and FtsA$_{E124A}$(*Geissler et al., 2003*) weakly bypassed the requirement of FtsN. These mutations also rescued the FtsL$_{L86F/E87K}$ mutant when overexpressed (*Karimova et al., 1998*). We thus examined whether similar mutations in FtsA of *A. baumannii* can bypass the need of Aeg1 by constructing substitution FtsA mutants D124A, V144L, and Q285W, which correspond to E124A, I143L, and R286W in its *E. coli* counterpart, respectively (*Figure 3—figure supplement 1*) and tested them in strain Δ*aeg1*(P$_{BAD}$::Aeg1). Whereas FtsA$_{Q285W}$ cannot detectably suppress Aeg1 depletion, expression of FtsA$_{D124A}$ or FtsA$_{V144L}$ partially resecured the growth of strain Δ*aeg1*(P$_{BAD}$::Aeg1) in medium without ara (*Figure 3D*). Despite multiple attempts, we were unable to detect the expression of these FtsA mutants by immunoblotting using antibody specific to the Flag epitope fused to these alleles (*Figure 3D*, lower panels), suggesting that the low activity exhibited by these mutants may result from low protein levels.

The observation that mutations in genes involved in cell division bypass the need of Aeg1 suggests that this protein participate in the same cellular process. We thus analyzed the cellular localization of Aeg1 in *A. baumannii* by introducing a plasmid that expresses Aeg1-mCherry from the P$_{BAD}$ promoter. Fluorescence microscopy analysis revealed that Aeg1 localized at the septum of dividing cells (*Figure 3E*), which validated the genetic evidence for its role in cell division.

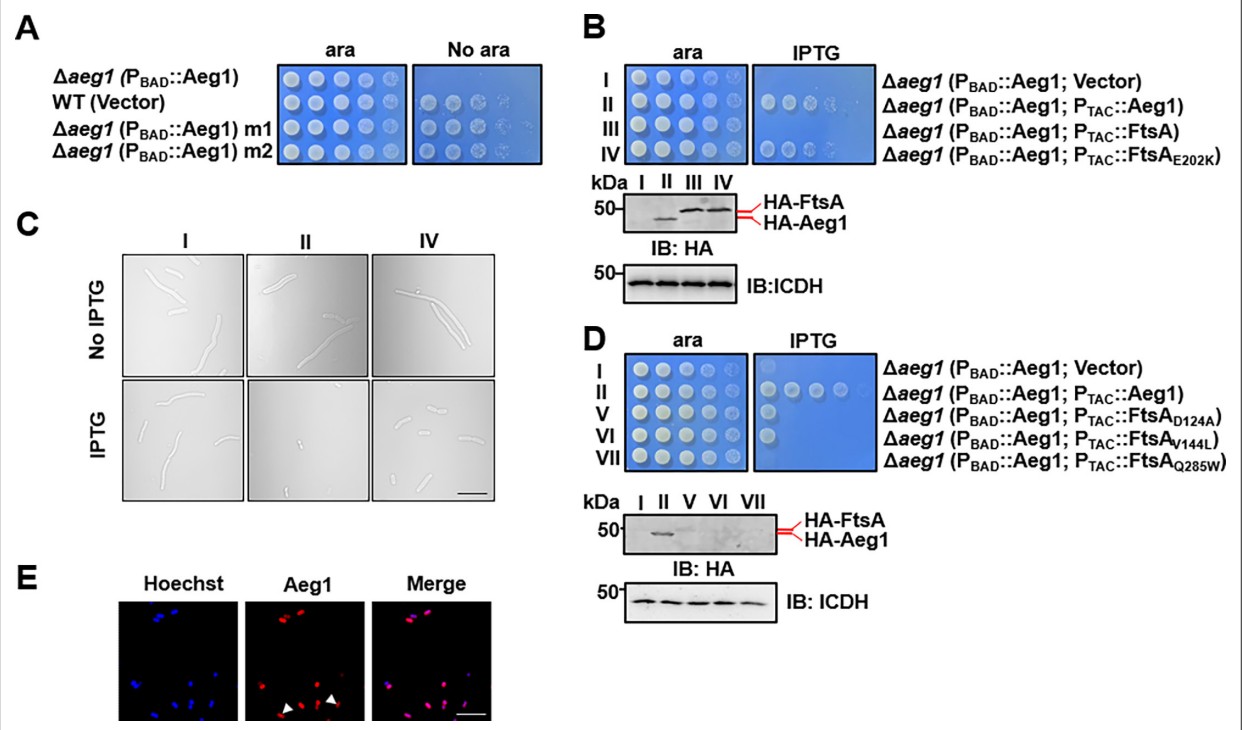

**Figure 3.** Mutations in proteins involved in cell division bypass the requirement of Aeg1. (**A**) Isolation of two suppressor mutants that gained viability in the absence of Aeg1. Cells serially diluted with water were spotted onto Luria Bertani (LB) agar with or without ara. Note that the Δ*aeg1*(P$_{BAD}$::Aeg1) strain (top row) cannot grow on medium without ara and the wild-type strain (second row) can grow on both conditions. Similar results were obtained in three independent experiments. (**B**) The FtsA$_{E202K}$ mutant suppressed the requirement of Aeg1. Cells of strains derived from Δ*aeg1*(P$_{BAD}$::*aeg1*) that harbored the vector (I), pHA-Aeg1 (II), pHA-FtsA (III), or pHA-FtsA$_{E202K}$ (IV) were spotted onto LB agar containing ara (left) or isopropyl-β-D-thiogalactopyranoside (IPTG; right), Images were acquired after 18 hr incubation at 37°C (**B**, upper panels). Expression of Aeg1, FtsA, or FtsA$_{E202K}$ in these strains were detected by immunoblotting with the HA-specific antibody after lysates of IPTG-induced cells being resolved by SDS–PAGE (Sodium Dodecyl Sulfate Polyacrylamide Gel Electrophoresis)). Isocitrate dehydrogenase (ICDH) was probed as a loading control (**B**, lower panels). Similar results were obtained in three independent experiments. (**C**) The suppression mutants assumed normal cell morphology. Bacterial strains derived from Δ*aeg1*(P$_{BAD}$::*aeg1*) grown in LB broth containing ara for 6 hr were diluted into fresh medium with the inducer and the cultures were induced with IPTG for 4 hr prior to being processed for imaging. Images were representatives of three parallel cultures. Bar, 10 µm. (**D**) Additional FtsA mutants bypassed the need of Aeg1. Mutant FtsA$_{D124A}$, FtsA$_{V144A}$, or FtsA$_{Q285A}$ expressed from the IPTG-inducible P$_{TAC}$ was introduced into strain Δ*aeg1*(P$_{BAD}$::Aeg1) and serially diluted cells of resulting strains were spotted onto LB agar supplemented with ara or IPTG. Images were acquired after incubation at 37°C for 18 hr (upper panels). Expression of Aeg1, FtsA$_{D124A}$, FtsA$_{V144A}$, and FtsA$_{Q285A}$ in these strains. Total protein of IPTG-induced cells resolved by SDS–PAGE and proteins transferred onto nitrocellulose membranes were detected by immunoblotting with the HA-specific antibody. ICDH was probed as a loading control (lower panels). Similar results were obtained in three independent experiments. (**E**) The Aeg1-mCherry fusion localizes to division constrictions. Wild-type *A. baumannii* harboring P$_{BAD}$::Aeg1-mCherry grown to the mid-log phase in LB broth containing 1% ara were processed for imaging. Bar, 10 µm. Images are representative of three parallel cultures.

The online version of this article includes the following source data and figure supplement(s) for figure 3:

**Source data 1.** PDF file containing original western blots for *Figure 3B, D*, indicating the relevant bands and treatments.

**Source data 2.** Original files for western blot analysis displayed in *Figure 3B, D*.

**Figure supplement 1.** Alignment of FtsA proteins between *A. baumannii* and *E. coli*.

## Aeg1 interacts with multiple core components of the *A. baumannii* divisome

To further explore the possibility that Aeg1 is a component of the *A. baumannii* divisome, we determined its interactions with known cell division proteins using the Cya-based bacterial two-hybrid method (*Tsang and Bernhardt, 2015*). Effective complementation between the fused T25 and T18 domains requires their localization to the cytoplasmic compartment. Because the predicted transmembrane topology of Aeg1 is not unequivocal, with one model suggesting an N-terminal transmembrane segment and another proposing a more centrally located membrane-spanning domain. To circumvent potential artifacts arising from the ambiguous topological models of Aeg1, we generated

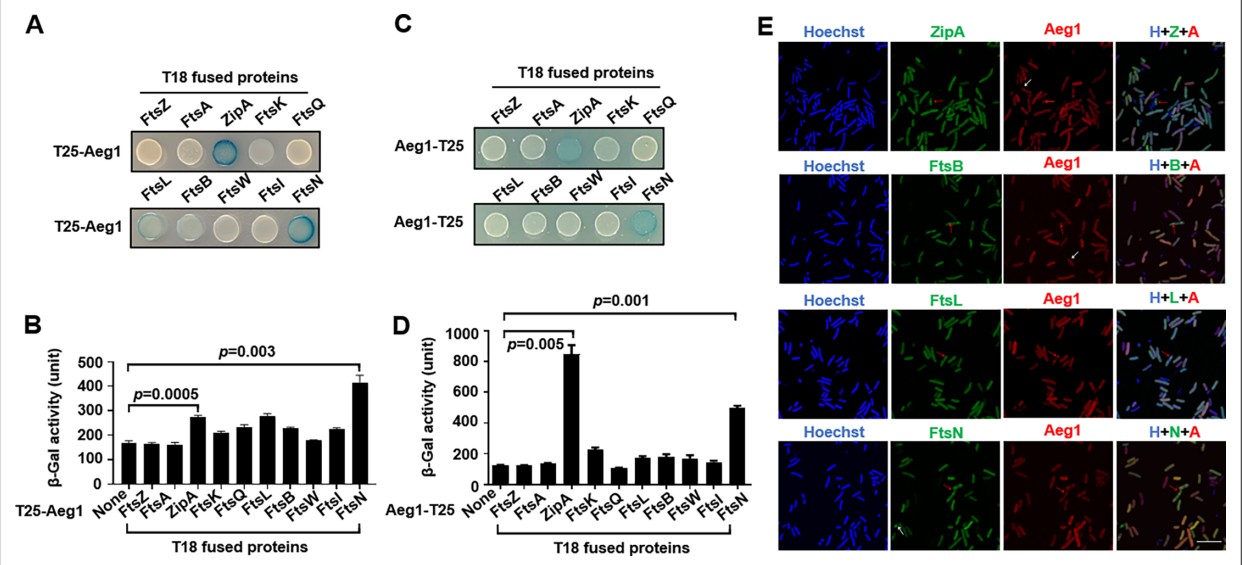

**Figure 4.** Aeg1 interacts with multiple core proteins of the *A. baumannii* divisome. (**A–D**) Derivatives of strain *E. coli* BTH101 containing T18-Aeg1 and T25 fusion of the indicated proteins were spotted onto Luria Bertani (LB) agar containing X-gal. T25 domain was fused to the N-terminus (**A, B**) or C-terminus (**C, D**) of Aeg1. Images were acquired after incubation for 18 hr at 37°C (**A, C**). Interactions determined by quantitative measurement of β-galactosidase activity (**B, D**). Results shown were from three independent experiments. Statistical analysis was performed by Student's *t*-test. (**E**) Aeg1 colocalized with a set of core divisome proteins. GFP(Green Fluorescent Protein). fusion of ZipA, FtsL, FtsB, or FtsN was expressed in strain WT(pJL03::mCherry-Aeg1). Saturated cultures of bacterial strains harboring the GFP and mCherry fusions were diluted into fresh LB broth. After the cell density of the subcultures reached until OD$_{600}$ = 0.6, isopropyl-β-D-thiogalactopyranoside (IPTG) (0.25 mM) and ara (0.25%) were added to induce the expression of fusion proteins for 4 hr prior to being processed for imaging. Red arrows indicate the sites of colocalization. Cells in which the proteins did not colocalize were shown by white arrows. Bar, 10 μm. Images are representative of three independent cultures.

The online version of this article includes the following figure supplement(s) for figure 4:

**Figure supplement 1.** GFP and mCherry fusion proteins rescued the growth defects of their corresponding mutants.

two distinct fusion constructs, tethering the T25 domain to either the N- or C-terminus of the Aeg1 protein. We then assessed the complementation between these Aeg1–T25 fusions and the potential binding partners fused to the T18 domain to evaluate the interaction between Aeg1 and other Fts proteins. Among the tested proteins FtsZ, FtsA, ZipA, FtsK, FtsQ, FtsL, FtsB, FtsW, FtsI, and FtsN, Aeg1 detectably interacts with ZipA, FtsK, FtsL, FtsB, and FtsN. Intriguingly, ZipA and FtsN appearing to have higher binding affinity with Aeg1 whether the T25 reporter domain was fused to the N- or C-terminus of the Aeg1 construct (*Figure 4A–D*), as indicated by the hydrolysis of X-gal by derivatives of the *E. coli* reporter strain BTH101 (*Tsang and Bernhardt, 2015*) harboring appropriate protein fusions. When the interactions were quantitatively determined by measuring galactosidase activity, the binding of FtsN or ZipA to Aeg1 appeared the strongest among all examined protein pairs (*Figure 4A–D*). This observation suggests that the precise topology of Aeg1 does not significantly impact its ability to engage these binding partners.

We next analyzed the localization of putative Aeg1-interacting proteins identified in our bacterial two-hybrid experiments. To this end, we coexpressed the mCherry-Aeg1 fusion with GFP fusion of ZipA, FtsL, FtsB, or FtsN in the WT strain from the ara- and IPTG-inducible promoter, respectively. Prior to localization studies, we first sought to examine the functionality of the GFP fusions by individually deleting the chromosomal gene of *zipA*, *ftsB*, *ftsL*, and *ftsN* in *A. baumannii* strains expressing the fusion proteins from a plasmid. Our results indicate that each GFP fusion was capable of complementing the growth defects caused by deleting the corresponding gene (*Figure 4—figure supplement 1A*). Similarly, we found that expression of the mCherry-Aeg1 fusion allowed the deletion of the *aeg1* gene (*Figure 4—figure supplement 1B*). In each case, the growth of the strain expressing the fusion protein complementing the chromosomal deletion was comparable to that of the wild-type strain (*Figure 4—figure supplement 1A*). These strains were then used in experiments aiming at investigating the localization of the relevant proteins. Saturated cultures of strains derived from WT(P$_{BAD}$::mCherry-Aeg1) (grown for 16 hr) that harbored the GFP fusions were diluted into fresh LB

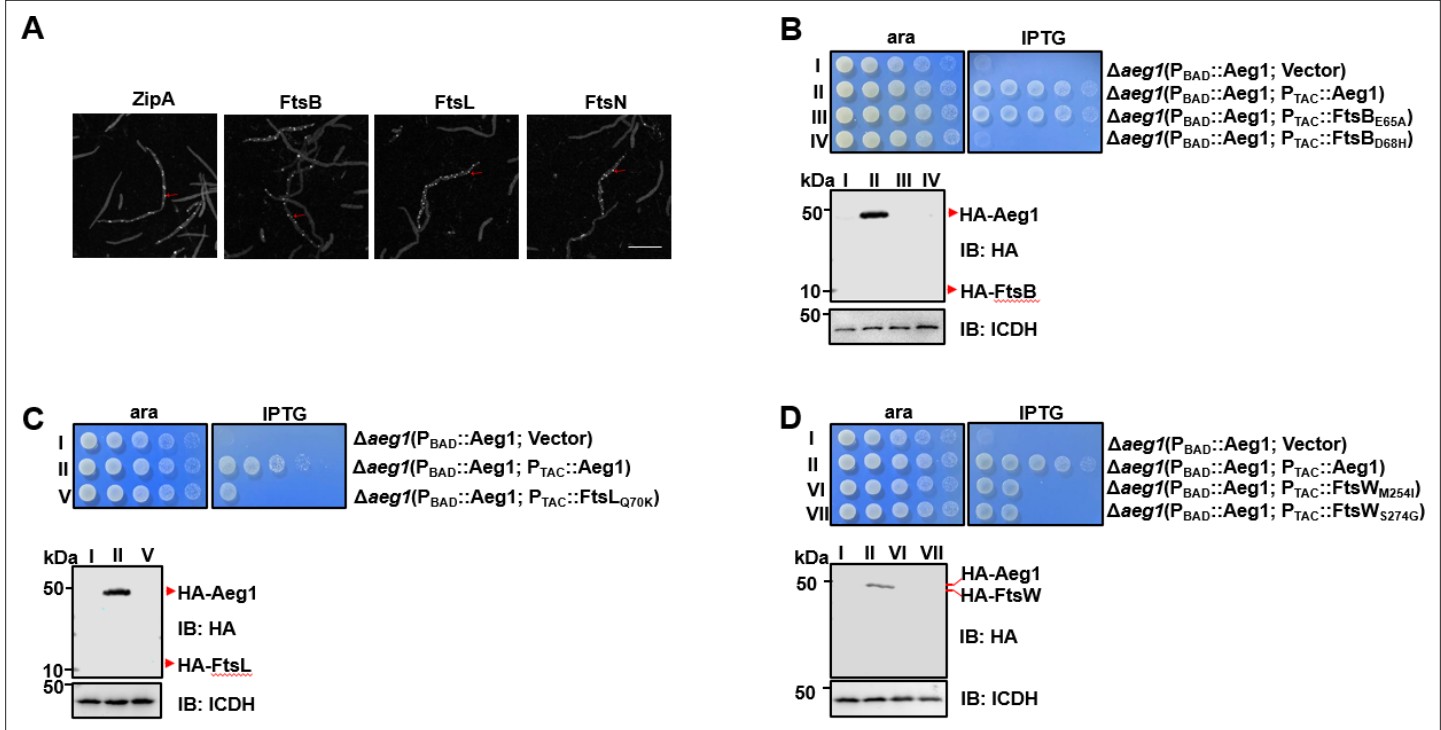

**Figure 5.** Aeg1 dictates the cellular localization of core divisome proteins. (**A**) Aeg1 deficiency prevented core divisome proteins from localizing to the middle cell. Cells of derivatives of strain Δ*aeg1*(pJL03::mCherry-Aeg1) expressing GFP fusion of each of the divisome proteins from the P$_{TAC}$ promoter grown in medium with ara were processed for imaging. The red arrows representative examples of the inability of fusion proteins to target to the septal ring. Bar, 5 μm. Images are representative of three parallel cultures. A few dominant active mutants of FtsB and FtsW bypassed the need of Aeg1 by *A. baumannii*. Mutant FtsB$_{E65A}$ (**B**), FtsB$_{D68H}$ (**B**), FtsL$_{Q70K}$ (**C**), FtsW$_{M254I}$ (**D**), or FtsW$_{S274G}$ (**D**) was expressed from the isopropyl-β-D-thiogalactopyranoside (IPTG)-inducible promoter P$_{TAC}$ in strain Δ*aeg1*(pJL03::Aeg1). Serially diluted cells were spotted onto Luria Bertani (LB) agar containing ara or IPTG. Images are acquired after incubation at 37°C for 18 hr (**B–D**). Similar results were obtained in three independent experiments. The expression of the protein was probed with the HA-specific antibody after IPTG induction (**B–D**). Isocitrate dehydrogenase (ICDH) was probed as a loading control.

The online version of this article includes the following source data and figure supplement(s) for figure 5:

**Source data 1.** PDF file containing original western blots for *Figure 5B–D*, indicating the relevant bands and treatments.

**Source data 2.** Original files for western blot analysis displayed in *Figure 5B–D*.

**Figure supplement 1.** Δ*aeg1*(P$_{BAD}$::mCherry) strain unable to exhibit expression of Aeg1-mCherry after 6 hr period without ara.

**Figure supplement 2.** Alignment of Fts proteins between *A. baumannii* and *E. coli*.

broth. When the cell density of the subcultures reached OD$_{600}$ = 0.6, IPTG (0.25 mM) or ara (0.25%) were added to induce the expression of fusion proteins for 4 hr and samples were processed for microscopic analysis. Consistent with results from the protein interaction experiments using the bacterial two-hybrid assay, we found that Aeg1 colocalized with ZipA, FtsL, FtsB, and FtsN (*Figure 4E*, red arrows), but occasional non-colocalization was also observed (*Figure 4E*, white arrows). Thus, Aeg1 interacts with multiple core cell divisome proteins of *A. baumannii*.

## Aeg1 is an essential core cell division protein in *A. baumannii*

The interaction map between Aeg1 and core divisome proteins prompted us to postulate that the absence of Aeg1 prevented these proteins from localizing to the midcell. To test this hypothesis, we introduced plasmids that direct the expression of GFP fusion of each of the divisome proteins into strain Δ*aeg1*(P$_{BAD}$::mCherry-Aeg1). We evaluated the protein of Aeg1-mCherry at 2, 4, and 6 hr after withdrawing arabinose and found that at the 2 and 4 hr time points mCherry-Aeg1 was still readily detectable (*Figure 5—figure supplement 1*). Importantly, we found that removal of arabinose for 6 hr rendered Aeg1-mCherry undetectable in approximately 90% of the cells. We thus used the 6 hr inducer withdraw to examine the effects of Aeg1 depletion. In cells depleted of Aeg1, all of the examined core division proteins displayed midcell mistargeting, including ZipA, FtsB, FtsL, and FtsN

(*Figure 5A*). These results suggest that Aeg1 is a new member of cell division protein, which plays a role in positioning several other core proteins at the septum of *A. baumannii*.

During the formation of the divisome, FtsN was believed to be the last protein being recruited to the septum where it converts FtsQLB and FtsA into their active forms, which subsequently activate the septal peptidoglycan synthase FtsWI (*Park et al., 2021*; *Park et al., 2020*). If the interaction between Aeg1 and FtsN is critical for the ability of the latter to switch on FtsQLB and FtsA, constitutively active mutants of FtsQLB or FtsA should suppress the need of Aeg1 for cell viability. To test this hypothesis, we expressed HA-FtsB$_{E65A}$ and HA-FtsL$_{Q70K}$ of *A. baumannii* (equivalent to mutants of their *E. coli* counterparts that allow the bypass of the requirement of FtsN) (*Figure 5—figure supplement 2*; *Park et al., 2021*; *Li et al., 2021*) in strain Δ*aeg1*(P$_{BAD}$::Aeg1) on medium without ara. Although the expression of HA-FtsB$_{E65A}$ was not detectable with the HA-specific antibody, induction of this mutant completely restored the growth of the Δ*aeg1* mutant in the absence of Aeg1 (*Figure 5B*). In contrast, a similarly constructed plasmid that directs the expression of HA-FtsB$_{D68H}$ cannot bypass the requirement of Aeg1 under the same conditions (*Figure 5B*). Similarly, expressed HA-FtsL$_{Q70K}$ detectably rescued bacterial growth without Aeg1 at a level considerably lower than that of FtsB$_{E65A}$ (*Figure 5C*). The low or undetectable activity exhibited by FtsL$_{Q70K}$ and FtsB$_{D68H}$ may result from the low protein levels in strain Δ*aeg1*(P$_{BAD}$::Aeg1). Nevertheless, these results support our hypothesis that Aeg1 functions upstream of FtsN, probably by functioning as its activator or recruiting factor.

It is known that expression of mutants FtsI$_{K211I}$, FtsW$_{M269I}$, and FtsW$_{E289G}$ driven by a strong promoter can bypass the need of FtsN in *E. coli* (*de Boer, 2010*). The ability of FtsB$_{E65A}$ and FtsA$_{E202K}$ to suppress the growth defect of the Δ*aeg1* mutant prompted us to examine whether constitutively active alleles of FtsW or FtsI have the same effects. Overexpression of FtsW$_{M254I}$ and FtsW$_{S274G}$ of *A. baumannii* detectably rescued the growth of the strain in which Aeg1 had been depleted (*Figure 5D*). FtsI from *A. baumannii* does not have a residue corresponding to K211 of its *E. coli* counterpart (*Figure 5—figure supplement 2*), we thus did not test the effect of this mutation. Therefore, our results demonstrate that Aeg1 may collaborate with FtsN to activate at least a part of FtsQLB and FtsA for subsequently activation of FtsWI during cell division. Together with results that the variant FtsA$_{E202K}$ suppressed the growth defect caused by *aeg1* depletion (*Figure 3B*), these observations support a model in which the Aeg1–FtsN complex is involved in the activation of FtsQLB and FtsA.

## Discussion

Powerful In-seq DNA sequencing technologies based on saturated transposon insertion libraries have allowed the identification of genes essential for bacterial viability under specific conditions, including commonly used rich media such as LB broth (*Wang et al., 2014*). Using a conditional gene deletion method, we have revealed that from 10 putative essential genes identified by this method, three previously annotated as hypotheticals are essential for *A. baumannii* viability, suggesting that assignment of essentiality to genes identified by methods of this kind needs to be individually verified.

Our assignment of *aeg1* as an essential gene is consistent with a recent study which revealed its essentiality (termed *advA*) by transposon insertion sequencing, which also observed cell elongation upon the depletion of this gene (*Pichoff et al., 2012*). Our results provided at least two additional lines of evidence to support the notion that Aeg1 is a core cell division protein in *A. baumannii*. First, random suppression mutations that bypassed the requirement of Aeg1 mapped to a classic cell division gene and mutations in some cell division genes known to circumvent the need of specific cell division proteins rescued the growth defect caused by Aeg1 depletion (*Figure 3*). Second, Aeg1 interacts with at least a subset of proteins involved in cell division (*Figure 4A–D*).

In the process of bacterial cell division, over 30 distinct proteins are assembled into a complex typically referred to as 'divisome' (*Haeusser and Margolin, 2016*). Ten of these are normally essential for both cell division and survival, and they are now mainly suggested to be the core of the divisome in various bacteria (*Haeusser and Margolin, 2016*; *Lutkenhaus, 2017*). These essential division proteins are mostly named Fts (filamentation temperature sensitive), which stems from the fact that inactivating one of these *fts* genes causes cells to become filamentous, which eventually failed to survive (*Haeusser and Margolin, 2016*; *Carson et al., 1991*). Among these, the FtsQLB complex not only serves as a scaffold for recruitment of subsequent divisome proteins (*Guzman et al., 1997*; *Craven et al., 2024*), but also plays an important role in regulating the activity of the divisome (*Park et al., 2021*; *Park et al., 2020*). The constriction control domain (CCD) of FtsL was previously defined as

a critical component for switching the conformation of FtsLB from an off to an on state in order to activate FtsWI by enabling the interaction between FtsI[peri] and the AWI (activation of FtsWI) region of FtsL (*Park et al., 2021*; *Park et al., 2020*; *Sánchez et al., 1994*). Division obstruction caused by Aeg1 depletion can be effectively circumvented by suppression mutations (FtsB_{E65A}) in its CCD domain and by activation mutations (M254I and S274G) of FtsW (*Figure 5D*), suggesting that Aeg1 functions upstream of these proteins.

In addition to FtsQLB-mediated activation of FtsI in the periplasm, the constitutively active form of FtsA (FtsA*) acting on FtsW in the cytoplasm, these two events functions synergistically to activate FtsWI for sPG synthesis (*Karimova et al., 1998*). FtsA is a member of the actin family (*van den Ent and Löwe, 2000*; *Szwedziak et al., 2012*), which interacts with FtsN via a unique IC domain positioned between the IA and IIA domains (*Baranova et al., 2020*; *Pichoff et al., 2015*; *Goehring and Beckwith, 2005*). Little is known about the function of the IA and IIA domains of FtsA. That the E202K mutation in the IIA domain bypasses the requirement of Aeg1 by *A. baumannii* (*Figure 3B*) suggests that Aeg1 and FtsA work together to activate FtsW.

The maturation of the core divisome is facilitated by the recruitment in a nearly sequential order of the late-division proteins FtsK, FtsQ, FtsL/FtsB, FtsW, FtsI, and FtsN (*Du et al., 2019*; *Addinall et al., 1997*; *Lyu et al., 2022*). Once the latest division protein, FtsN, is recruited, division-specific peptidoglycan synthesis conferred by the FtsWI complex is activated to build the septum (*Lyu et al., 2022*; *Weiss, 2015*). More recent studies suggest that by binding to FtsA, a small amount of FtsN is recruited to the division site earlier than previously thought (*Park et al., 2021*; *Goehring and Beckwith, 2005*; *Piehaud and Second, 1951*) where it induces a conformational switch in both FtsA and the FtsQLB subcomplex on both sides of the membrane (*Park et al., 2021*). Furthermore, binding between FtsA

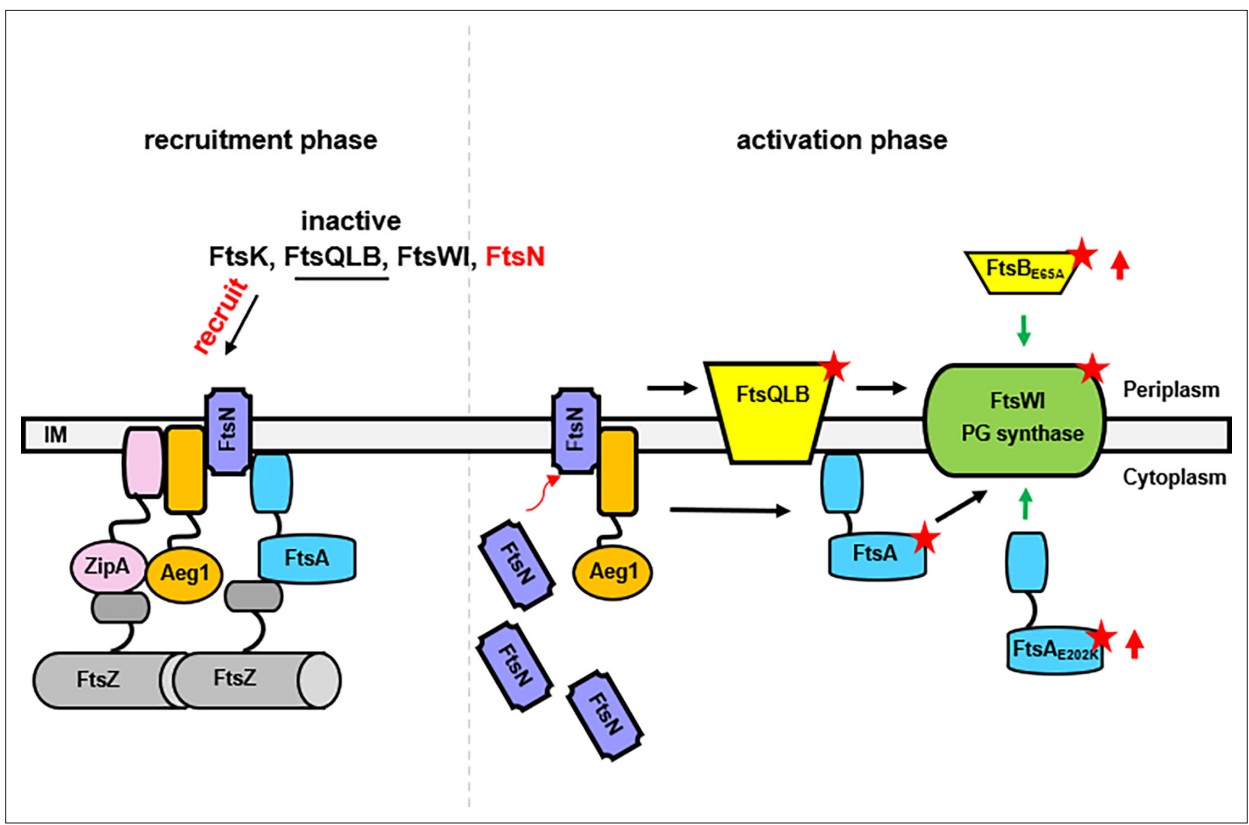

**Figure 6.** A model for the function of Aeg1 in cell division of *A. baumannii*. In the early stages of Z-ring assembly, anchor proteins FtsA and ZipA tether FtsZ to the inner face of the plasma membrane. In the initial phase of cell division, a small amount of FtsN is recruited to the divisome in a FtsA-dependent manner (left, recruitment phase) (*Wroblewska et al., 2007*). The Aeg1–ZipA–FtsN complex acts as a dynamic scaffold for recruiting downstream of Fts proteins. Once more FtsN is recruited to the divisome, it will activate both FtsA and the FtsQLB complex, which in turn activates FtsI in the periplasm (*Li et al., 2021*). Note that constitutively active form of FtsA (FtsA*) acts on FtsW in the cytoplasm (*Karimova et al., 1998*) so did the hyperactive mutants FtsA_{E202K}. These mutants all bypassed the need of Aeg1 (right, activation phase). Constitutively active mutants were indicated by a star. The red arrows next to FtsA_{E202K} or FtsB_{E65A} indicate overexpression of these mutants.

and FtsN is regulated by ZipA (*Goehring and Beckwith, 2005*). Our observation of strong binding of Aeg1 to ZipA and FtsN (*Figure 4A–D*) prompted us to propose a model in which Aeg1 participates in cell division by anchoring to the division site together with ZipA where it recruits FtsN, which in turn signals to both the cytoplasmic and periplasmic branches of peptidoglycan synthesis involving FtsA and the FtsQLB complex, respectively (*Park et al., 2021*; *Figure 6A*). The placement of Aeg1 as one of the early arrivals in the hierarchical sequential assembly of the divisome is consistent with the fact that the requirement of Aeg1 can be bypassed by constitutively active mutants of FtsA$_{E202K}$ (*Figure 3*), FtsB (FtsB$_{E65A}$), FtsL (FtsL$_{Q70K}$), and of FtsW (FtsW$_{M254I}$ and FtsW$_{S274G}$) (*Figure 5*).

It is worth noting that that although homologs of Aeg1 are present in many Gram-negative bacteria, including some recent isolate of *E. coli*, it appears to be absent in more extensively studied model strains such as K12 (*Figure 1—figure supplement 2*). It is possible that in strain K12, its activity is conferred by a yet unidentified protein that employs a conserved mechanism in cell division. Future study aiming at elucidating the biochemical and structural basis of the activity of Aeg1 and its homologs, and their relationship with other components of the divisome may provide not only insights into the mechanism of bacterial cell division but also leads for the design of antibiotics with novel mechanism of action.

## Materials and methods
### Bacterial strains, growth conditions, and plasmids construction
All bacterial strains and plasmids used in this study are listed in *Supplementary file 3*. All *A. baumannii* strains were derivatives of strain ATCC17978 (*Yildirim et al., 2016*). Bacteria were grown in LB broth or agar. Antibiotics were added to media when necessary. For *E. coli*, the concentrations used were: ampicillin, 100 μg/ml, kanamycin, 30 μg/ml, chloramphenicol, 30 μg/ml, and gentamicin, 10 μg/mL. For *A. baumannii*, kanamycin, 30 μg/ml, gentamicin, 10 μg/ml, and streptomycin, 100 μg/ml.

To construct pUT18C-Flag, two oligos with sequences coding for the Flag tag and appropriate restriction sites were mixed at a 1: 1 ratio, heated to 100°C for 5 min and allowed the mixed DNA to anneal by gradually cooling down to room temperature (RT). The resultant DNA fragment was then inserted into *SalI*/*ClaI* digested pUT18C (*Tsang and Bernhardt, 2015*). We introduced an *Eco*RV site into the inserted DNA fragment to facilitate the identification of plasmids carrying the intended insert. To make pJL05 that allows P$_{TAC}$-driven expression of HA-tagged protein, primers containing the coding sequence of the HA tag were used to amplified the promoter region of pJL02 (*Jie et al., 2021*), PCR product was inserted as a *PciI*/*BamH*I fragment. This manipulation replaced the Flag tag on pJL02 with an HA tag. pJL03 is a pVRL1-based plasmid harboring P$_{BAD}$, which allows expressing Flag-tagged proteins in *A. baumannii* (*Jie et al., 2021*).

The DNA fragment were amplified from genomic DNA of *A. baumannii* strain and digested with appropriate restriction enzyme (*Supplementary file 3*), following insert into similarly digested plasmids. The experimental procedure entails fusion of the GFP gene with the target gene, followed by ligation of the resulting construct into the pJL05 vector for subsequent expression.

### Introduction of plasmids into *A. baumannii*
Plasmids were introduced into *A. baumannii* by electroporation (*Finan et al., 1986*) or by triparental mating. Electrocompetent cells were prepared using a previously described protocol (*Jhjeimg, 1972*) with minor modifications. Briefly, saturated cultures of *A. baumannii* were diluted 1:100 into 50 ml LB medium to grow for 18 hr at 37°C and cells were harvested by centrifugation (4000 × *g*) at RT for 5 min. Cells were washed twice with 10 ml of 10% glycerol, resuspended in 1 ml of 10% glycerol, cells aliquoted in 50 μl. were stored at −80°C. For transformation, 1 μg of plasmid DNA was mixed with 50 μl competent cells and electroporation was performed in 0.2 cm cuvettes on a Bio-Rad Gene Pulser system (2.5 kV/cm, 25 μF, 200 Ω).

The *E. coli* MT607 harboring pRK600was used for triparental mating. Briefly, 100 μl of overnight cultures of the donor, recipient, and the helper strain were mixed and cells were harvest by centrifugation (4000 × *g*) for 5 min. Mixed cells washed twice with 1 ml of sterile water were spotted onto LB agar and the mating was allowed to proceeded for 4–6 hr at 37°C before plating onto selective medium.

## Construction of in-frame deletion mutants of *A. baumannii*

To delete the putative essential genes, we first introduced a plasmid expressing the gene of interest into *A. baumannii* and the resulting strains were used to construct chromosomal deletion mutants using a method based on the suicide plasmid pSR47s (*Merriam et al., 1997*). Briefly, the coding region of the gene was inserted into pJL03, which allows expression of Flag-tagged protein from the P$_{BAD}$ promoter (*Jie et al., 2021*) and the resulting plasmid was introduced into *A. baumannii* by electroporation. Strains in which the chromosomal deletion is complemented by GFP or mCherry fusions of the relevant Fts proteins were similarly constructed.

To construct the plasmid used for in-frame deletion of *A. baumannii* genes, 1.0 kilobase DNA fragment was amplified from the upstream and downstream of the gene to delete, respectively, digested with appropriate restriction enzymes and inserted into similarly digested pSR47s (*Merriam et al., 1997*). The primers were designed so that in each case the gene to be deleted was replaced with an open reading frame consisting of the first and last 15 residues linked by two amino acids encoded by the sequence of the restriction enzyme used to fuse the two DNA fragments (*Supplementary file 3*). The plasmid was then introduced into the *A. baumannii* strain harboring the pJL03 derivative expressing the gene to be deleted from the P$_{BAD}$ promoter by triparental mating. Transconjugants obtained on LB agar containing 1% ara were grown in LB broth containing 5% sucrose for 18 hr prior to being streaked onto LB agar supplemented with 5% sucrose. Deletion mutants were identified by colony PCR with primers corresponding to the 5′ end of the upstream fragment and the 3′ of the downstream fragments.

## EMS mutagenesis

The *A. baumannii* strain Δ*aeg1*(P$_{BAD}$::Aeg1) was grown in LB broth containing 1% ara to saturation, cells from 2 ml culture collected by centrifugation were washed twice with cold buffer A [60 mM K$_2$HPO$_4$, 33 mM KH$_2$PO$_4$, 7.6 mM (NH$_4$)$_2$SO$_4$, 1.7 mM sodium citrate, pH = 7]. Cells resuspended in 2× buffer A supplemented with 1.4% EMS were incubated in a shaker (220 rpm) at 37°C for 15 min for 30, 45, and 60 min, respectively. Treated cells harvested by centrifugation were washed twice with buffer A prior to being plated onto LB agar supplemented with 5% glucose. Colonies appeared were purified and tested by spotting diluted cells onto LB agar with or without ara, respectively, to confirm the ability to grown in the absence Aeg1. To eliminate the possibility that the ara-independent growth phenotype was due to mutations in the P$_{BAD}$ promoter, the plasmid from each mutant candidate was rescued and the promoter region was sequenced. Only mutants harboring an intact P$_{BAD}$ promoter were retained for further study.

Genome sequencing was performed by Suzhou Golden Wisdom Biological Technology Co, Ltd, and mutations were identified using the genome of the parent strain ATCC 17978 as reference (*Smith et al., 2007*).

## Bacterial two-hybrid assays

To assess interactions between Aeg1 and *A. baumannii* Fts proteins, we employed the *Bordetella pertussis* adenylate cyclase (Cya)-based bacterial two-hybrid system (*Tsang and Bernhardt, 2015*). The coding region of Aeg1 was fused to the T25 fragment of Cya and testing Fts proteins were individually fused to the T18 fragment on pUT18C-Flag, which allows the detection of the fusion protein with the Flag-specific antibody (*Supplementary file 3*). Interaction was evaluated in the cya⁻ *E. coli* strain BTH101 (*Tsang and Bernhardt, 2015*) by the ability of the testing strains to hydrolyze X-Gal. Quantitative measurement of β-galactosidase activity with performed with ortho-nitrophenyl-β-galactoside (ONPG) using an established protocol (*Busiek and Margolin, 2014*). Testing bacterial strains were cultured in 2 ml LB broth containing 0.5 mM IPTG for 18 hr, and 0.2 ml cells were harvest and resuspended in 1 ml Z-buffer (60 mM Na$_2$HPO$_4$, 40 mM NaH$_2$PO$_4$·H$_2$O, 1 mM MgSO$_4$, 10 mM KCl, and 7 ml of 2-mercaptoethanol per liter). After adding 50 μl 1% (m/v) SDS and 100 μl of chloroform, cells were lysed by 30 s vortex and 50 μl of ONPG (40 mg/ml) was added. The reaction was terminated with 500 μl of 1 M Na$_2$CO$_3$ after ONPG hydrolysis had occurred as determined the presence of the yellow color product. β-Galactosidase activity is defined as in Miller units using the following equation: (OD$_{420}$ × 1000)/[(time of incubation in min) × (volume of cell suspension in ml) × (cell density of the culture in OD$_{600}$)]. All assays were done in triplicate.

## Antibodies and immunoblotting

HA-specific antibody (1:1000) was purchased from Santa Cruz (Cat #sc-7392), and rabbit anti-ICDH (1:2000) was described previously (*Jie et al., 2021*). To prepare protein samples, we diluted (1:100) saturated bacterial cultures into fresh LB broth. When $OD_{600}$ of the cultures reached 0.6, IPTG was added to a final concentration of 0.5 mM. Approximately $1 \times 10^9$ cells were collected, resuspended in 100 µl SDS sample buffer and were boiled for 5 min. Proteins separated by SDS–PAGE were transferred onto nitrocellulose membranes, which were first blocked with 5% nonfat milk for 1 hr followed by incubation with appropriate primary antibodies for 14 hr. Washed membranes were incubated with the appropriate IRDye-labeled secondary antibodies and signals were detected using an Odyssey CLx system (LI-COR).

## Fluorescence imaging

To determine the cellular localization of Aeg1 and its colocalization with members of the divisome, we constructed pJL03::Aeg1-mCherry, which expressed the fusion from the $P_{BAD}$ promoter (*Guzman et al., 1995*). GFP chimera of Fts proteins were expressed from pJL05 driven by the $P_{TAC}$ promoter inducible by IPTG. Each of the GFP-Fts fusion was introduced into strain WT($P_{BAD}$::mCherry-Aeg1).

To visualize the localization of the fusions, bacterial strains were grown to saturation in LB broth containing ara and the cells washed twice with PBS were diluted to a density of approximately $5 \times 10^6$ cells/ml. The cultures were split into two 2 ml subcultures, the one used to determine how the absence of Aeg1 impacted the cellular localization of Fts proteins received only IPTG but not ara. Cultures were incubated in a shaker for an additional 6 hr, and 100 µl cells were withdrawn and fixed by incubation in glutaraldehyde for 20 min at RT. After washing twice with PBS, samples were stained with Hoechst 33342 (1:5000) (Invitrogen, cat#H3570) and mounted on slides for observation with an Olympus IX-83 fluorescence microscope.

## Bioinformatics analysis

Aeg1 homologs were identified using tBLASTn (*Gertz et al., 2006*) using the Aeg1 protein (*A1S_3387*) from strain ATCC17978 (GenBank: CP000521.1) as the query sequence. Clustal omega (https://www.ebi.ac.uk/Tools/msa/clustalo/) was used to perform multiple sequence alignments. The homology comparison of Fts proteins between *E. coli* and *A. baumannii* was also performed using Clustal omega. Jalview was used for editing and viewing sequence alignments (*Waterhouse et al., 2009*). The transmembrane regions of Aeg1 were predicted by the SMART Web sites (*Käll et al., 2004*) and Phobius web server (*Käll et al., 2007*; *Geisinger et al., 2020*).

## Data quantitation, statistical analyses

Student's *t*-test was used to compare the mean between two groups each with at least three independent samples.

## Acknowledgements

This work in part was supported by the Natural Science Foundation of Jilin Province grant YDZJ202201ZYTS025 (XC), Jilin Science and Technology Agency grant 20200403117SF (LS), Science and Technology Development Project of Changchun City 23YQ10 (LS), and The Bethune Project of Jilin University 2024B20 (LS).

## Additional information

### Funding

| Funder | Grant reference number | Author |
|---|---|---|
| Natural Science Foundation of Jilin Province | YDZJ202201ZYTS025 | Xiao Chu |

| Funder | Grant reference number | Author |
|---|---|---|
| Jilin Science and Technology Agency grant | 20200403117SF | Lei Song |
| Science and Technology Development Project of Changchun City | 23YQ10 | Lei Song |
| The Bethune Project of Jilin University | 2024B20 | Lei Song |

The funders had no role in study design, data collection, and interpretation, or the decision to submit the work for publication.

## Author contributions

Xiao Chu, Data curation, Supervision, Writing - original draft, Project administration; Lidong Wang, Data curation, Formal analysis, Methodology; Yiheng Zhu, Data curation; Zhengshan Feng, Investigation, Methodology; Qingtian Guan, Data curation, Investigation; Lei Song, Resources, Project administration; Zhaoqing Luo, Supervision, Funding acquisition, Writing – review and editing

## Author ORCIDs

Xiao Chu ⓘ https://orcid.org/0009-0007-0779-6688
Lei Song ⓘ https://orcid.org/0000-0002-4115-065X
Zhaoqing Luo ⓘ http://orcid.org/0000-0001-8890-6621

None https://doi.org/10.7554/eLife.87922.4.sa1
Author response https://doi.org/10.7554/eLife.87922.4.sa2

# Additional files

## Supplementary files

• Supplementary file 1. The essential genes identified and analyzed within this study.

• Supplementary file 2. A comparative analysis of substitution and insertion–deletion (InDel) mutations found in strains M1 and M2 relative to the genome sequence of *A. baumannii* strain ATCC 17978.

• Supplementary file 3. A comprehensive list of the bacterial strains, plasmids, and primers utilized throughout the study.

• MDAR checklist

## Data availability

All analyses are included in the manuscript and supporting files, and no files are stored in other databases.

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
