## [Editor Report · eLife assessment]

This **useful** study shows that the essential Acinetobacter baumannii gene Aeg1 likely plays an key role in cell division. The strength of the work is the discovery that the depletion of Aeg1 leads to cell filamentation and that gain-of-function mutations in cell division genes FtsB and FtsL rescue the lethality of Aeg1 depletion. However, Aeg1's localization pattern and its requirement for other division proteins' localizations require further characterization of the functionality of fluorescent fusion proteins, fluorescence images of higher quality, and improvements in statistic qualifications, leaving the study' evidence for Aeg1's exact role in cell division **incomplete** at this time. In conclusion, the critical role of Aeg1 in the assembly of the A. baumannii divisome has yet to be established unambiguously.

---

## [Referee Report · None]

In this study, the authors confirm that one of the genes classified as essential in a Tn-mutagenesis study in A. baumannii, Aeg1, is, in fact, an essential gene. The strength of the work is that it discovered that the depletion of Aeg1 leads to cell filamentation and that activation mutations in various cell division genes can suppress the requirement for Aeg1. These results suggest that Aeg1 plays an important role in cell division. The work's weakness is that it lacks convincing evidence to define Aeg1's place or role in the divisome assembly pathway. It is unclear whether proteins are at the division site under the wildtype condition and when Aeg1 is depleted, and whether Aeg1 is indeed required for a set of division proteins to the division site.

Reviewer comments:

The revised manuscript partially addressed two of the three major concerns from the previous assessment: (1) the functionality test of fluorescent fusion proteins using a spotting assay, and (2) membrane protein topology in the bacterial two-hybrid assays by constructing a C-terminal T25 fusion.

(1) In the spotting assay, all fluorescent fusion proteins rescued the growth of the corresponding deletion strain, which suggests these fusion proteins are functional. However, fluorescent images of these fusion proteins were diffusive, and only a few cells showed the expected midcell/membrane localization pattern for cell division proteins. This observation raised the concern that these fusion proteins may be cleaved in the middle, leading to the separation of the untagged fusion partner and diffusive fluorescent protein in the cytoplasm, which would explain the positive spotting rescue results. This phenomenon is commonly observed in other bacterial species. A western blot using an antibody targeting either the fluorescent protein or the fusion partner is widely used to examine whether the fusion protein is expressed at its full length.

(2) The authors constructed a C-terminal fusion of Aeg1 and showed that it still interacted with ZipA and FtsN. This result supports the authors' suggestion that the N-terminus of Aeg1 may not be the predicated membrane-targeting domain. Along the same line, the membrane topology of ZipA should also be considered. ZipA's N terminus is in the membrane facing the periplasm, and its C terminal domain is in the cytoplasm. Therefore, the PUT18C fusion will place the T18 domain of ZipA in the periplasm. All other division proteins' N termini are in the cytoplasm.

(3) Colocalization images did not show significant midcell localizations for each fluorescent protein; most cells showed diffusive cytoplasmic fluorescence. In all other species, midcell localization of cell division proteins is prominent in dividing cells, especially for early division proteins such as ZipA (at least 40-50% of cells show midcell bands). In A. baumannii, divisome localization timing may differ from other species, but this possibility needs to be established before the colocalization pattern is examined. Compounding this issue is that in Aeg1 depletion strains, some cells expressing ZipA, FtsB, FtsL, and FtsN fusions showed roughly regularly spaced puncta in long filamentous cells. It is hard to explain why this was observed if, under the WT condition, these fusions do not localize to the midcell. These results again raised concerns that these fusion proteins may not be functional and the observations are protein aggregates.

Besides these major issues, experimental observations did not support some claims in the main text. For example: (1) In the two-hybrid assay, only ZipA and FtsN showed significant interactions with Aeg1, as judged by the darkness of the blue spots. FtsL and FtsB showed pale spots. The quantified values accompanying this figure did not appear to agree with the image. (2) The spotting rescue assay showed that only FtsB-E56A and FtsA-E202K was able to bypass Aeg1 depletion (full dilution set comparable to that of Aeg1 complementation), but the main text claimed that FtsA-D124A and V144L, and FtsW-M254I and S274G also rescued the growth. These claims could be misleading.

---

## [Author Response]

The following is the authors’ response to the previous reviews.

(1) The reviewers asked to clarify the BTH assay: The fused T25 and T18 domains must be in the cytoplasmic to complement successfully. The authors stated that the N terminus of Aeg1 transverses the membrane once, which means that the T25-Aeg1 will have T25 in the periplasm. However, T18C vector fusion with other division proteins will have T18C of ZipA in the periplasm (ZipA's N terminus is on the periplasmic side of the inner membrane) while that of FtsN in the cytoplasm (FtsN's N terminus is in the cytoplasm). As such, it isn't easy to understand why T25-Aeg1 showed positive results for both ZipA and FtsN. Note that FtsL, FtsB, and FtsI all have the same topology as FtsN but showed negative results. It is possible that these fusion proteins do not fold correctly, and hence, the results cannot be interpreted directly. The authors did not address this concern but only cited that BTH is a commonly used assay for protein-protein interactions.

In response to the editor's comments and the concerns raised by the reviewer, we have performed two sets of Aeg1-T25 fusion experiments to determine whether the Aeg1 topology impacts protein interactions measured by bacterial two-hybrid (BTH) assays. In the first set of experiments, we fused the T25 domain to the N-terminus of Aeg1 and still observed strong binding of Aeg1 to ZipA and FtsN, respectively. Similar results were obtained from the second set of experiments in which the T25 domain was fused to the C-terminus of Aeg1.

These results indicate that the precise topology of Aeg1 does not significantly impact its ability to engage these binding partners. Aeg1 is predicted to harbor a single transmembrane domain, however, the precise location of this transmembrane segment differs in predictions made by different algorithms. The SMART Web site (1) predicted the transmembrane region to be located at the N-terminus of Aeg1 (7-29 aa). In contrast, Phobius, based on HMM （2， 3）suggested the transmembrane segment is situated more centrally within the Aeg1 protein (134-151 aa), and further proposed that the N-terminus may function as a signal peptide. This latter prediction also provides a potential explanation for the larger-than-expected molecular weight of the Aeg1 truncation mutant observed in the Western blot shown in Fig 1C. The removal of the putative signal peptide may have altered the protein structure, affecting its electrophoretic mobility. As a result, we are more inclined to favor the topology model for Aeg1 predicted by Phobius.

(2) It is still difficult to identify the midcell localization patterns of Aeg1 and other division proteins from microscopy images (Fig. 4C and Fig. 5A). In Fig 4C, only ZipA and Aeg1 formed clear, regular band-like colocalization patterns. Others formed irregular co-localized puncta along the cell length, different from the expected midcell localization patterns. Cells also appeared to be much longer than WT cells, suggesting cell division defects. The most likely reason for these aberrant localization patterns and filamentous cells is that GFP/mCherry-fusions of these division proteins are not functional and become dominant negative, interfering with proper cell division. The authors need to test the functionality of these fusion proteins before they can be used for imaging. (The authors also mislabeled Hoechst and the division protein GFP panels labels in this figure.)

Thank you for raising this important point. To examine the functionality of the fluorescence protein fusion constructs, we have painstakingly performed conditional knockout of the genes of interest (*zipA*, *ftsB*, *ftsL*, and *ftsN*) in *A. baumannii* strains inducibly expressing the corresponding fusion protein. We found that these fluorescence protein fusions were able to fully rescue the growth of the mutant lacking the corresponding *fts* gene (Figure 4-figure supplement 1). Concurrently, we have also successfully knocked out the *aeg1* gene under conditions *in trans* expression of an mCherry-Aeg1 fusion protein, which was able to effectively rescue the growth defects of the Δa_eg1_ mutant (Figure 4-figure supplement 1). We then introduced the functional fluorescence protein fusions into wild-type cells and observed the co-localization of Aeg1 with the relevant Fts proteins. The results showed that Aeg1 indeed co-localized with ZipA, FtsB, FtsL, and FtsN (Fig.4E, red arrows), but occasional non-co-localization was also observed (Fig.4E, white arrows).

We have utilized the functional fluorescence protein fusion constructs to analyze the localization of relevant Aeg1-interacting proteins in the Δ*aeg1* strain upon Aeg1 depletion. Our results showed that the depletion of Aeg1 indeed impacted the midcell localization of the several Aeg1-interacting Fts proteins.

References

(1) Letunic I, Khedkar S, Bork P. SMART: recent updates, new developments and status in 2020. *Nucleic acids research*. 2021;49:D458-d60.doi: 10.1093/nar/gkaa937.

(2) Käll L, Krogh A, Sonnhammer EL. A combined transmembrane topology and signal peptide prediction method. *Journal of molecular biology*. 2004;338:1027-36.doi: 10.1016/j.jmb.2004.03.016

(3) Käll L, Krogh A, Sonnhammer EL. Advantages of combined transmembrane topology and signal peptide prediction--the Phobius web server. *Nucleic acids research*. 2007;35:W429-32.doi: 10.1093/nar/gkm256